# Workplace-Based Organizational Interventions Promoting Mental Health and Happiness among Healthcare Workers: A Realist Review

**DOI:** 10.3390/ijerph16224396

**Published:** 2019-11-11

**Authors:** Patricia Gray, Sipho Senabe, Nisha Naicker, Spo Kgalamono, Annalee Yassi, Jerry M. Spiegel

**Affiliations:** 1School of Population and Public Health, University of British Columbia, Vancouver, BC V6T 1Z3, Canada; patricia.gray@alumni.ubc.ca (P.G.); annalee.yassi@ubc.ca (A.Y.); 2Gauteng Department of Health, Gauteng Provincial Government, 45 Commissioner Street, Marshall Town (Johannesburg) 2147, South Africa; sipho.senabe@gauteng.gov.za; 3National Institute of Occupational Health, National Health Laboratory Service, Braamfontein, Johannesburg 2001, South Africa; NishaN@nioh.ac.za (N.N.); SpoK@nioh.ac.za (S.K.); 4School of Public Health, University of Witwatersrand, Parktown 2000, South Africa; 5Department of Environmental Health, Faculty of Health Sciences, University of Johannesburg, Johannesburg 2028, South Africa

**Keywords:** mental health, occupational mental health, healthcare workers, mental health promotion

## Abstract

Mental illness, deemed globally to account for 32% of years lived with a disability, generates significant impacts on workplaces. In particular, healthcare workers experience high rates of mental ill health such as burnout, stress, and depression due to workplace conditions including excessive workloads, workplace violence and bullying, which also produces negative effects on patients as well as on the happiness and wellbeing of those who remain at work. This review was undertaken to synthesize the evidence on workplace-based interventions at the organizational level promoting mental health and wellbeing among healthcare workers, to identify what has been receiving attention in this area and why, especially considering how such positive effects are produced. A search of three premier health-related databases identified 1290 articles that discussed healthcare workers, workplace interventions, and mental health. Following further examination, 46 articles were ultimately selected as meeting the criteria specifying interventions at the organizational level and combined with similar studies included in a relevant Cochrane review. The 60 chosen articles were then analyzed following a realist framework analyzing context, mechanism, and outcome. Most of the studies included in the realist review were conducted in high-income countries, and the types of organizational-level interventions studied included skills and knowledge development, leadership development, communication and team building, stress management as well as workload and time management. Common themes from the realist review highlight the importance of employee engagement in the intervention development and implementation process. The literature review also supports the recognized need for more research on mental health and happiness in low- and middle-income countries, and for studies evaluating the longer-term effects of workplace mental health promotion.

## 1. Introduction

Mental illness is estimated to globally account for 32.4% of years lived with a disability [1] and to generate significant impacts on workplaces with depression and anxiety disorders costing US$1 trillion dollars in lost productivity in 2017 [2]. As evidence of the growing recognition of mental ill health in the workplace, in May 2019, the World Health Organization for the first time classified burnout as an “occupational phenomenon” in the eleventh revision of the International Classification of Diseases [3]. There is also an expanding awareness of not only absence and the direct costs due to mental ill health, but also the effects on workers who remain on the job. Employers are increasingly paying attention to presenteeism—decreased productivity due to health problems—by employees who remain present at work [4]. Indeed, presenteeism has been shown to contribute a larger economic cost than absenteeism and employer health costs [5]. With the economic burden so high, the relative returns from investing in mental health are favorable: every dollar invested in scaling up treatment for common mental illnesses such as depression and anxiety leads to a four-fold return in better health and ability to work [6].

In the workplace, there are multiple factors recognized to be determinants of workers’ mental health [7]. These include high job demand, low job control, low workplace social support, effort-reward imbalance, low organizational procedural justice, low organizational relational justice, organizational change, job insecurity, temporary employment status, atypical working hours, bullying, and role stress [8]. In addition, non-work determinants such as family status and social support networks are also important predictors of workers’ mental health [9]. The healthcare workplace, in particular, experiences high rates of mental illness such as burnout, stress, post-traumatic stress disorder, anxiety, and depression [10,11,12,13,14] due to workplace conditions such as excessive workloads [15,16,17,18,19,20], working in emotionally-charged situations [21], stigma against seeking care [22], and workplace violence [23,24], among other factors. Mental ill health of healthcare workers has additionally been associated with an increased risk of patient safety incidents, poorer quality of care due to low professionalism, and reduced patient satisfaction [25]; medical errors [26,27]; quality of care, patient falls, medication errors, and infections [28]; lower patient satisfaction [29]; and patient safety outcomes [30]. Poor mental health of healthcare workers further affects healthcare systems by intensifying shortages among the workforce as a result of a reduction in work effort or those leaving the practice [16,31] as well as the significant economic costs this generates [32].

Over and above the substantial burden of mental illness and ill health, the value of positive mental health and wellness has been increasingly recognized and captured through constructs such as happiness, an approach receiving growing international attention such as by the United Nations’ High-Level Meeting on Wellbeing and Happiness in 2012. This orientation conforms well with the World Health Organization’s definition of health as “a state of complete physical, mental and social wellbeing and not merely the absence of disease or infirmity” [33]. For the purposes of this review, the term “mental health” herein encompasses both positive and negative conceptions of mental wellbeing.

Although there has been an increase in mental health promotion and prevention programs globally, only 7% of such initiatives are workplace-based [34]. Indeed, the Global Happiness Policy Report (2018) calls for more research to expand the causal evidence base on work and wellbeing, and to evaluate workplace interventions promoting worker wellbeing [35]. In 2015, a Cochrane systematic review evaluated evidence on the effectiveness of interventions to prevent occupational stress in healthcare workers [36], but was restricted to studies measuring work-related stress and/or burnout by using validated tools. As such, application of a more holistic definition of mental health including a broader scope of mental health outcomes such as the psychosocial work environment and satisfaction is warranted to appreciate the potential benefits of improving work environments. Furthermore, the Cochrane review included only quantitative studies meeting stringent conditions: randomized controlled trials for individual-level interventions, controlled before and after studies, and interrupted time series for organizational-level interventions excluding cross-sectional and qualitative studies that could contribute to better understanding the effectiveness of interventions in various situations.

To explore the practical challenges of understanding *how* workplace-based organizational interventions work effectively in multidimensional and unavoidably diverse healthcare contexts, the realist review method provides a more nuanced approach for ascertaining what works for whom, in what circumstances, in what respects and how [37], rather than definitively assessing the efficacy of a relatively standard intervention producing a discrete outcome. This orientation is well suited to address the recognized need for greater focus on the process of organizational interventions including the why and how of successful (and unsuccessful) interventions [38]. While we note that a realist review is currently being undertaken to address mental illness in physicians [39], there is a need to equally take stock of the impacts of interventions on nurses, midwives, and other healthcare professionals as well as other healthcare support staff. Accordingly, in examining the question of how can workplace-based organizational interventions improve the mental health and wellbeing of healthcare workers, we elected to conduct a realist review that takes into consideration (a) a wider definition of outcomes of interest; (b) diverse study designs worthy for inclusion in realist analyses; and (c) a comprehensive range of healthcare workers.

## 2. Methods

### 2.1. Search Strategy

In our exploratory review of literature relevant to our question of interest, we surveyed research in the areas of global mental health, mental health and the workplace, mental health and the healthcare workplace, mental wellbeing and happiness, happiness and the workplace, workplace-based mental health interventions, and workplace-based mental health interventions for healthcare workers. A search of MEDLINE, CINAHL (*Cumulative Index to Nursing and Allied Health Literature*), and PsycINFO was conducted in November 2018 (see Appendix B: Search Terms). MEDLINE was selected based on its extensive and premier coverage of health and biomedical research. CINAHL was selected to additionally capture studies related to nursing and allied health professionals. PsycINFO was selected for its command of psychology, which is particularly relevant for the subject area of the review: mental health. All three databases were searched for journal articles published since the inception of the respective databases (i.e., there were no restrictions placed on the dates of coverage).

### 2.2. Selection and Appraisal of Documents

The database search (MEDLINE, CINAHL, and PsycINFO) initially yielded 1496 articles. After duplicates were removed, 1290 articles remained. The titles and abstracts of the 1290 articles were then screened, primarily looking at the target population (healthcare workers), intervention (workplace-based), and outcome (mental health including positive concepts of mental health). A broad scope of “mental health” was intentionally adopted including positive constructs such as psychological resilience, quality of life, life satisfaction, and happiness as well as negative constructs such as stress, burnout, and mental disorders (more specific inclusion and exclusion criteria for the full selection and appraisal process is listed below in Table 1). Based on the title and abstract review, 1012 articles were excluded, leaving 278 articles for full-text assessment. Following the full-text review, 177 articles were excluded (see Figure 1). The remaining 101 articles were then classified by the category of workplace-based intervention by using the three categories from the Cochrane review: cognitive-behavioral, relaxation, and organizational, which were developed by the authors of the Cochrane review as there is no major framework available for categorizing preventive stress interventions [36]. As this realist review is focused on organizational-level interventions, the articles categorized as cognitive-behavioral and relaxation were excluded (*n* = 55). Then, the studies on organizational interventions included in the Cochrane review were added (*n* = 21), which included all of the articles included in the organizational category even if they were not included in the Cochrane review’s meta-analysis. Then, duplicates (*n* = 4) and studies on students and trainees (*n* = 2) were removed. In addition, an article that did not cover the implementation of an intervention was removed. This left the final set of articles to include for the synthesis (*n* = 60). Five of the 60 were articles on the same study(ies) as other article(s), and therefore their analysis was combined to represent one study/intervention. Thus, this review comprises 55 unique studies represented in 60 research articles (listed in Table 2; a full summary of the included articles is available in Appendix A). The specific inclusion and exclusion criteria used for the selection of articles is as follows.

### 2.3. Data Extraction

General demographic and characteristic information extracted from the selected studies were country where the study took place; type of healthcare worker (nurse, physician, etc.); type of healthcare workplace (hospital, residential care facility, etc.); and type of specialized care, if applicable (e.g., intensive care unit, emergency medicine, etc.). These characteristics were selected to provide a summary of the locations and participants who have been included in studies to date as well as to provide an overview of the contexts for the studies.

Each study was summarized using a realist framework comprising a brief description of the context including why and how the intervention occurred; a brief description of the underlying mechanism for how the intervention was intended to work; and, a brief description of the outcome. These were selected to facilitate the analysis and synthesis of the studies using the realist framework of context–mechanism–outcome.

Methods of each study were also summarized comprising a brief description of the intervention itself; the type of study design; length of the follow-up period, if applicable; the construct/measure(s) of mental health used in the study; and, whether or not there was an improvement in the construct/measure(s) of mental health. These were selected to provide a summary of the research methods used and to compare the types of interventions studied and methods used. The wide range of mental health outcome constructs/measures used in the selected studies—intentionally included to encompass a broad definition of mental health and wellness—precluded the extraction of quantitative measures of outcomes.

Finally, any other findings and/or relevant notes that the reviewer thought were significant for the purposes of this realist review were noted, in particular findings and observations related to the realist framework of context–mechanism–outcome.

### 2.4. Analysis and Synthesis Processes

The analysis and synthesis followed an iterative process. The selected articles were read and reviewed by the primary reviewer, during which time the general demographics and characteristics of each study (as listed above; location of study, type of healthcare workers, type of work environments, etc.) were documented and summarized across the selection of studies (see Document Characteristics below). The three elements of the realist framework (context, mechanism, outcome) as well as a brief description of the intervention were extracted from each of the articles by the primary reviewer in note form. For the context, the impetus for the intervention and/or a brief description of the initial state of the workplace was noted if available in the article (e.g., hospital restructuring, high absenteeism, promising initial evidence, etc.). Contextual factors that were mentioned to have aided, hindered, or otherwise influenced the intervention were also noted. For the mechanism, the underlying strategy, theory and/or functionality of the intervention was noted (e.g., reducing work hours, enhancing work-related skills, etc.).

In order to facilitate summaries and comparisons across studies, the mechanisms were then also categorized into the following types, based on the common themes found in the collection of studies: skills and knowledge development, leadership development, communication and team building, stress management, or workload and time management. If the intervention comprised more than one type of mechanism, it was categorized as “mixed”. This was done to facilitate synthesis across the variety of interventions in the selected studies.

A second review of the articles was then done to extract the following information from each study: study design, length of follow-up period (if applicable), and measure(s) of mental health. During the second review, the reviewer also reviewed the information and notes from the first review.

The elements of the realist framework from the articles were then analyzed thematically, with a focus on process-related factors across the studies. The diversity of both mental health constructs and types of interventions precluded the direct comparison or evaluation of interventions amongst one another, therefore, the focus was placed on a thematic analysis related to patterns of mechanisms for developing and implementing workplace-based interventions to promote mental health among healthcare workers.

## 3. Results

### 3.1. Document Characteristics

Table 2 summarizes basic characteristics of all 55 relevant studies in 60 articles [40,41,42,43,44,45,46,47,48,49,50,51,52,53,54,55,56,57,58,59,60,61,62,63,64,65,66,67,68,69,70,71,72,73,74,75,76,77,78,79,80,81,82,83,84,85,86,87,88,89,90,91,92,93,94,95,96,97,98,99] selected in our review, with a consolidated description of this material following.

Most of the studies (*n* = 51) were from high-income country contexts with the remaining four studies from upper middle-income countries, as categorized by the World Bank [100]. The continent most represented in the studies was Europe (*n* = 22), followed by North America (*n* = 21), Asia (*n* = 6), and Oceania (*n* = 5). Only one study was from Africa. Half of the studies (*n* = 32) were published from 2010 onward. The earliest study was published in 1993.

Most of the studies targeted one type of healthcare worker, with the most common being nursing staff. Of the studies targeting one particular type of healthcare worker, nurses and nursing staff was the most common type of healthcare worker (*n* = 25), followed by physicians (*n* = 5). Twenty-one studies included more than one type of healthcare worker such as nurses, midwives, physicians, administrative staff, and/or managerial staff.

More than half of the studies (*n* = 32) emphasized the unique characteristics of certain healthcare work environments such as residential care facilities and intensive care units, for example, noting the particular demands and emotionally-charged nature of the work environments. Of the studies explicitly focusing on a particular type of healthcare service, the most common was elder and/or adult residential care (*n* = 11), followed by intensive care (*n* = 6), mental health/psychiatric care (*n* = 5), oncology (*n* = 4), and emergency medicine (*n* = 2).

A variety of designs were used in the included studies. The most common study design was quasi-experimental (*n* = 17), followed by a randomized trial, either a cluster randomized trial or an individual randomized trial (*n* = 15). Other methods used were longitudinal surveys (*n* = 11), mixed methods (*n* = 4), cross-sectional surveys (*n* = 4), and qualitative methods (*n* = 4). Of the four mixed methods studies, two comprised longitudinal surveys and focus groups, one used a retrospective pre/post survey and interviews, and one used longitudinal and qualitative surveys. The four qualitative studies all used interviews, with one of the four also using a collective case study method in addition to interviews.

Among the quasi-experimental studies, randomized trials, longitudinal studies, and the three mixed method studies with a longitudinal component (*n* = 46), the duration of follow-up time ranged from 20 days to 10 years. Eight studies had a follow-up time of less than six months, fourteen studies had between 6–10 months, and nineteen studies had between 12–24 months. There was one study each with the follow-up times of three years, four years, seven years, and 10 years. One study did not indicate the duration of its follow-up time.

There were a variety of constructs of mental health and happiness used in the studies, with most studies (*n* = 41) using more than one construct. The most common construct was burnout (*n* = 27), followed by stress (*n* = 19), and job or work satisfaction (*n* = 14). Other constructs used in the studies include distress (*n* = 6), depression (*n* = 4), psychosocial work environment (*n* = 3), psychological wellbeing (*n* = 2), anxiety (*n* = 2), psychosomatic symptoms (*n* = 2), affect (*n* = 1), and resilience (*n* = 1), among others.

The types of interventions most commonly used were skills and knowledge development (*n* = 13), followed by communication and team building (*n* = 10), workload and time management (*n* = 9), stress management (*n* = 7), and leadership development (*n* = 3). Thirteen studies used mixed types of interventions (according to this categorization of mechanisms).

Twenty-five studies found an improvement in their measure of mental health. Seventeen found an insignificant or partial improvement including an improvement on only one dimension of a scale, for example. Three studies found an improvement in the short term, but it was not sustained in the long term. Ten studies found no improvement or a decline in their measure of mental health.

### 3.2. Main Findings

Several themes and patterns of mechanisms for developing and implementing interventions relevant to a realist framework emerged from the analysis.

#### 3.3.1. Stakeholder Engagement and Support

The importance of and necessity for stakeholder engagement has been highlighted repeatedly in multiple studies. For example, in their process evaluation, Uchiyama et al. [95] found that continuously engaging with key people such as unit leads was necessary for the intervention’s successful implementation. Another study highlighted potential issues arising from conflicting levels of support from different levels of the organization. In this case, the study was supported by the hospital administration who compensated the nurses for participation; however, it was not supported by all of the unit-level managers, who effectively barred participation through how they scheduled their staff [53]. This includes engagement and support across all levels of the organization, from management to frontline workers. In the introduction of a new telemedicine service, Romig et al. recognized that positive staff perceptions to the new technology would be key to its successful implementation so they showed the staff the benefits at the outset to help ensure uptake and usage of the technology [88]. Several studies noted the participatory nature of the interventions—involving frontline staff in the development and/or implementation of the interventions—as a key success factor [41,43,44,46,49,54,55,57]. The benefits of a participatory process were even further emphasized by studies that found the participation process itself as having a positive impact on employees [62,75].

#### 3.3.2. Approaches to Developing Interventions

Common approaches to identify which workplace factors to target in the interventions included initial engagement with the healthcare workers (e.g., initial baseline study to find context-specific factors affecting mental health), building on promising earlier research (e.g., following a pilot study), and/or based on a theory of mental health in the workplace. Some of the studies explicitly included the theoretical foundation of their intervention, either in combination with local needs or as the sole basis for a particular intervention target. This, for example, [43,44] targeted four theory-grounded psychosocial job factors: psychological demands, decision latitude, social support, and effort-reward imbalance, while Canadian research on improving civility in the work environment was based on the theoretical model of interpersonal relationships at work [65,66,67].

#### 3.3.3. Managing Expectations

A few studies noted the challenges and implications related to managing expectations around workplace mental health promotion activities. For example, Uchiyama et al. noted the lack of improvement in measures of the psychosocial work environment could be due to the higher expectations that some employees had based on the issues they were facing in the workplace, for instance, employees facing issues of workload and compensation may not have felt that interventions addressing team meeting scheduling and communication met their expectations for workplace improvements [41]. Ultimately, managing expectations could make the difference between successful and unsuccessful implementation. For example, Aust et al. identified the mismatch between employees’ expectations and program delivery as one of the reasons why the intervention “failed” (the psychosocial work environment at the hospital worsened by their measurement after the intervention) [42]. Conversely, elevated expectations for the interventions could have the opposite effect, inflating the true benefits of the mental health promotion activities based on employees’ positive expectations and hope for the change. For example, Bryan et al. found that the nurses’ “excitement for change” likely contributed to the observed short-term increase followed by a subsequent decrease in job satisfaction [45].

#### 3.3.4. Complexity of Evaluating Organizational-Level Interventions

A theme that emerged from the studies was the complexity of evaluating organizational-level interventions, in particular due to factors external to the study that can impact the outcome. For example, Proctor et al. [82] found no change in psychological distress among the intervention group, however, there were organizational and managerial changes that happened to occur at the same time as the study. The control group increased in psychological distress during that time, suggesting that the intervention was successfully able to mitigate the distress caused by the organizational changes, although purely from the results of the study, this would not necessarily have been apparent.

#### 3.3.5. Process vs. Mechanism

Several studies have addressed the challenge of distinguishing between the process of developing, introducing, and/or implementing an intervention and its impact on the mental wellbeing of employees, as opposed to the impact of the actual intervention on mental wellbeing. For instance, Bunce et al. also measured process variables and concluded that the process did in fact affect the outcome; indeed, the improvements in occupational stress following an interactive training program regressed to their initial levels one year later, also suggesting the immediate improvements were based on the process of engaging in the training program rather than the skill and knowledge garnered from the program [46]. Two considerations for future intervention development could be derived from this matter of process and mechanism. First, process matters. If the process itself could impact and influence the outcome, then it is important to be intentional and selective with the process. Second, if the process impacts the outcome, this could affect the longevity and subsequent measures of the success of an intervention among a workforce.

#### 3.3.6. Sustainability and Longevity of Interventions

Only two studies evaluated the longer-term effects of policy changes: one from California found that new legislation on minimum staffing levels for licensed nurses in hospitals increased job satisfaction [92], and one from Germany found that a policy limiting hospital physicians’ weekly working time led to no improvement in physicians’ mental health after ten years [86]. Most studies evaluated the effect of a discrete initiative, which begs the question of the sustainability and longevity of the effects following the completion of the study. As an exception, [58] noted that a facilitator was employed by the organization to continue the program following the close of the study. Three studies in particular found a short-term improvement following the intervention, which was not sustained in the long term [45,46,53].

#### 3.3.7. Broad Definition of Mental Health

Although the most commonly-used measure of mental health was burnout (see Document Characteristics above), there was a variety of constructs used in the studies including stress, work satisfaction, distress, depression, psychosocial work environment, psychological wellbeing, anxiety, psychosomatic symptoms, affect, and resilience. Even among the studies using burnout, many discussed differences in the three dimensions of emotional exhaustion, depersonalization, and personal accomplishment [101]. The multitude of constructs both reinforces the multifaceted nature of mental health in the workplace as well as advances the conceptualization of mental health beyond merely the presence or absence of mental disorders to also include more holistic measures of wellbeing and happiness.

## 4. Discussion

In line with the realist synthesis approach, the focus of this discussion is on contextual factors and processes of conducting interventions that influenced how certain mechanisms generated positive outcome [102]. Due to the complexity of factors influencing healthcare workers’ mental health, the broad definition of mental health including positive mental wellbeing such as happiness as well as the complexity of implementing and evaluating workplace-based interventions, a realist review method is well-suited to explore why and how interventions in this area work. In considering this here, we also reflect on areas where additional attention is needed.

The first theme that emerged was the importance of aligning the underlying reason, strategy, and/or theory with the structure and content of the intervention itself as well as with the mental health constructs that define success; this requires careful consideration of the specific needs of each population and the context and nuances of the design of the intervention. LaMontagne et al. have presented a framework for an integrated approach to workplace mental health interventions. This brings together “harm prevention” addressing workplace organization primary prevention initiatives, “positive mental health promotion” addressing individuals’ resilience in mitigating effects, and illness management through diagnosis, treatment, and reintegration [103]. These three approaches correspond with the traditional domains of public/occupational health, organizational development/psychology, and psychiatry. Although seemingly intuitive, it is important that consideration is given to how the targeted upstream organizational-level factors could affect change in individuals’ mental health and wellbeing, and the influences on the optional approaches being considered for implementation [103]. This also directly calls into question the positionality of those engaged in making these judgments, to ensure that the scope of possible interventions is not unduly restricted by the preferences of one particular stakeholder group.

The second theme was the importance of the engagement of employees across the organization; lack of engagement from employees or a certain group of employees was often cited as a reason why an intervention did not succeed or did not achieve the outcome anticipated. This theme has been strongly recognized to be of critical importance in workplace health promotion interventions [104,105]. Furthermore, the psychosocial safety climate (PSC)—a construct encompassing policies, practices, and procedures protecting the psychological health and safety of workers [106]—is positively correlated with employees’ engagement and job satisfaction, and negatively correlated with mental ill health [107], reinforcing the positive effects of participation and engagement across all levels of an organization. Similarly, workplace culture can impact the outcomes and success of work-based health promotion activities, particularly in the context of implementation. Beyond the specific programs and activities, an organization’s “culture of health”—fully integrating health into the organizational culture of how people think and act—is crucial for workplace-based health promotion, with elements that contribute to a culture of health including a physically-supportive environment, socially-supportive environment, leadership support, supportive middle management, peer encouragement and team building, and employee involvement and engagement [108]. Several studies have used participatory approaches in the development and/or implementation of the intervention. The theme of engagement also highlights the importance of providing employees with the time and capacity to participate, which may involve ensuring management support to allow employees to participate. Meaningful and repeated engagement can also help to manage expectations for the project so as to avoid the potential negative effects of unmet or mismatched expectations.

A third theme relates to managing complexity. There are the many factors at the individual, organizational, and societal levels that affect mental health in the workplace, and make it difficult to select a particular target on which to act as well as the difficulty in evaluating the effect of a specific change. Indeed, several studies have mentioned unrelated organizational changes that occurred at the same time as the intervention such as restructuring or layoffs that subsequently diminished or negated the effects of the intervention. The consideration of complexity also includes diversity within groups of workers with different needs, challenges, and factors that are contributing to their mental health at any given time, and therefore the difficulty in identifying and implementing an intervention that will work for a group of employees. The heterogeneity of most employee populations also presents challenges with consistent evaluation; for instance, a meta-analysis of workplace health promotion programs found larger effect sizes in younger populations [109]. This also highlights the importance of process considerations as inextricably linked to the mechanism for change. The complexity of occupational health interventions in healthcare is reflected in the recognition of particular challenges and considerations involved in the evaluation of such interventions such as the importance of context and suitability of research methods [110,111].

The fourth theme is the sustainability and longevity of both the intervention and the effect on employees’ mental health. For example, the three studies that specifically found a short-term improvement in workers’ mental health, but no long-term improvement, reflect the idea that a short-term initiative can improve mental health in the short term through an “excitement of change” effect while resources and attention are focused on the initiative in the short term [45]. In this regard, the same type of bias noted in the Hawthorne effect may be at play, reflecting an effect of attention being given more than the efficacy of the action put into effect [112]. On the other hand, the implementation of the intervention during a discrete time interval may underestimate potential effects if it does not provide sufficient exposure for the employees to the intervention. In addition, what is of particular importance is that the improvement may not be sustainable once the intervention ends. Within the often complex environment of healthcare, making incremental changes within a comprehensive transformation strategy has been identified as a guiding principle for ways to engage in the process of culture change [113], underscoring the importance of continuous improvement rather than discrete and singular efforts. Finally, another related consideration is who is responsible for the continued implementation of the intervention. For example, is it the workers’ continued responsibility to enact what had been introduced through group-based training and support intervention offered by the organization?

Occupational mental health is most commonly conceptualized from a pathological standpoint, characterized by the presence or absence of mental illness or disorders, rather than the presence of positive mental health and wellbeing. Indeed, in a qualitative study on job-related wellbeing, stress and burnout among healthcare workers in rural Ethiopia, most participants expressed a view of wellbeing as absence of stress rather than as a positive state [114]. Although there is increasing attention toward concepts of positive mental wellbeing including happiness in other disciplines such as economics [115,116,117,118], there remains an opportunity to integrate more positive dimensions and conceptualization of mental health in workplace-based interventions. The term “happiness” as an indicator of positive mental health can refer both to moods and emotions as well as to more long-term wellbeing and life satisfaction. The Organization for Economic Co-operation and Development defines subjective wellbeing as encompassing three elements: life evaluation (“a reflective assessment on a person’s life or some specific aspect of it”), affect (feelings or emotions, usually at a point in time), and eudaimonia (sense of meaning and purpose, or “psychological flourishing”) [118]. As for measuring happiness and mental wellbeing, life satisfaction is seen as a more reliable measure of overall wellbeing as it depends more on the continuing circumstances of people’s lives [115,119]. Therefore, measures of happiness based on life satisfaction are deemed better suited to capture longer-term and international differences in policies and institutions [116]. For instance, the World Happiness Report—an annual report published by the United Nations since 2012 on global happiness including a ranking of countries’ happiness levels—uses life satisfaction as its measure of happiness [120].

It is of critical importance to note that the large burden of mental illness is often exacerbated by stigma and discrimination. The negative impacts of stigma occur, for example, through victimization, mistreatment, loss of support networks, and difficulties accessing housing [121]. In addition, stigma has been found to deter or delay help-seeking among individuals with mental illness, with a disproportionate effect among ethnic minorities, youth, men, military personnel, and health professionals [122]. Stigma and discrimination can in turn worsen the social and economic costs of mental illness. Individuals who experience discrimination against mental illness in a healthcare setting have almost double the costs of health service usage [123]. In addition to the impact of stigma against mental illness, the stigma against infectious diseases such as HIV/AIDS and tuberculosis can in turn negatively impact people’s mental health [124]. Furthermore, in Low and Middle Income Countries (LMICs) experiencing the burden of both infectious and chronic non-infectious diseases including mental illness, the resulting intersection of stigmas can result in a syndemic [125].

As the themes discussed above all call into question the importance of attention to the mechanisms and processes for carrying out interventions to improve mental health and wellbeing, it is worth reflecting on the limited attention to the existence of “health and safety management” processes that have been introduced to promote the involvement and participation of workers representatives along with management and health professionals [126]. Perhaps this is a mechanism that should receive additional attention and be subject to evaluation itself.

Finally, the value of applying a realist-informed approach to the question of what works in promoting mental health and wellbeing in the workplace merits consideration. For example, while the 2015 Cochrane systematic review on preventing occupational stress in healthcare workers concluded that there was low-quality evidence that changing work schedules reduced stress, and other organizational-level interventions led to minimal to no changes in stress, [36], our review also found mixed evidence of workload and time management-related interventions affecting mental health. A strength of our review was the broad inclusion criteria for healthcare workers including nurses, midwives, physicians, social workers, aged care staff, and support staff. With the increasing move toward interdisciplinary and team-based healthcare, studying the impact on a diverse set of healthcare workers is essential, acknowledging the unique challenges and opportunities presented in each discipline.

This review furthermore highlights the need for research in this area from LMICs, particularly from African countries. With increasing demand for health human resources in LMICs, there is a need for evidence on how best to support the mental health of healthcare workers in these countries.

This review also highlights the need for robust definitions and approaches to mental health in the workplace, extending beyond the traditional focus on the absence of negative mental health to more broadly encompass constructs of mental wellbeing and happiness including long-term subjective wellbeing and life satisfaction.

### Limitations

A major limitation of the review is the lack of generalizability due to the variety of mental health constructs included. The variety of mental health constructs intentionally included in this review encompassed a “mental health” that was not solely the absence or presence of a mental illness or condition; while expanding on previous reviews in so doing, the range of outcomes and measures make direct comparisons across studies difficult. Second, the focus on only organizational-level interventions meant the exclusion of cognitive-behavioral and relaxation interventions. Although this selection criterion was intentional, this categorization of interventions may implicitly or explicitly impose a mutual exclusivity among types of interventions, when a holistic approach to mental health promotion in the workplace may be warranted. Third, the exclusion of grey literature from the literature search strategy may have precluded consideration of additional relevant research [89] conducted by organizations such as occupational health and safety agencies in the analysis. Fourth, there was only one reviewer who conducted the literature search, analysis, and synthesis process, which may have led to bias in both the selection and synthesis processes.

## 5. Conclusions

While the healthcare workforce experiences high rates of mental ill health, generating significant effects not only for the workers themselves but also on patients, there is limited evidence on how to promote mental health and wellbeing in the healthcare workforce. The collection of studies included in this review highlights the complexity of factors at the organizational level that influence mental health and the work environment.

There is strong rationale for approaching mental health promotion in the workplace from a continuous improvement perspective. This is due to the variety of factors influencing mental health in the workplace at any given time, regardless of the complexity of evaluating and measuring progress in this area. Moreover, attention to this challenge have been quite uneven globally. Recommendations for research on organizational-level interventions to promote mental health particularly include the need for more research in low- and middle-income countries. Additionally, there exists great opportunity for better integration of positive mental health and wellbeing constructs such as happiness in the context of workers’ mental health in intervention studies, so that efforts to improve this can be more strongly considered in future knowledge synthesis reviews.

## Figures and Tables

**Figure 1 ijerph-16-04396-f001:**
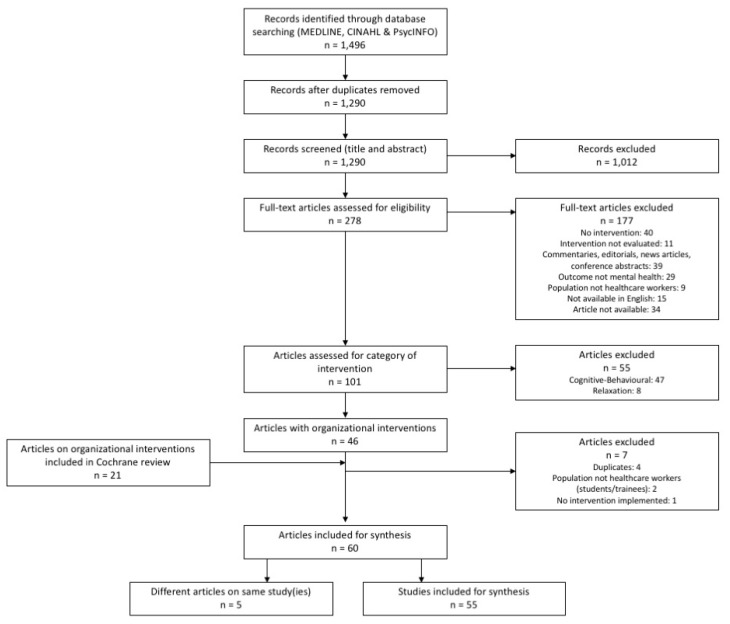
Flow diagram of search and selection process.

**Table 1 ijerph-16-04396-t001:** Summary of the inclusion and exclusion criteria.

Study Characteristic	Inclusion Criteria	Exclusion Criteria
Population	Healthcare workers (e.g., physicians, nurses, allied health professionals, etc.)	Medical or other health professional students or trainees (e.g., medical residents) Patients Family caregivers Veterinarians
Intervention	To promote mental health, defined broadly as both presence/absence of mental illness such as stress or anxiety, as well as positive concepts of mental health such as quality of life and life and/or job satisfaction	Studies on healthcare workers’ perceptions of interventions and/or programs generally (i.e., no specific intervention implemented) Interventions to reduce factors that may impact mental health such as workplace violence or bullying Interventions to reduce substance use Interventions to reduce mental health-related stigma Evaluation of return to work programs Tertiary interventions (e.g., clinical treatment of depression or other mental health condition; return to work programs for employees on sick leave due to depression)
Context	Interventions delivered in the workplace at the organizational level	Interventions delivered outside the workplace
Outcome	At least one measure of mental health (illness or wellness)	Studies with measures of work environment factors only (e.g., communication and collaboration among employees)
Study Design	Longitudinal observational study designs, including retrospective and prospective studies Experimental study designs, including randomized control trials and cluster randomized control trials Qualitative study designs Pilot studies	Models (theoretical or statistical) on ways to improve mental health
Publication	Published in a peer-reviewed journal Published since database inception (i.e., no minimum publication date)	Reviews, study protocols, editorials, letters to the editor, commentaries, theses or dissertations Grey literature

**Table 2 ijerph-16-04396-t002:** Summary of studies.

Citation	Reference number#	Country	Type of Health Worker	Workplace	Measure(s) of Mental Health
Ali et al. (2011)	[40]	United States	Physicians	Hospital	burnout, stress, work–home life imbalance
Arnetz and Hasson (2007)	[41]	Sweden	Nurses	Elder care organizations	psychosocial work environment
Aust et al. (2010)	[42]	Denmark	Mixed	Hospital	psychosocial work environment
Bourbonnais et al. (2006 and 2011)	[43,44]	Canada	Mixed	Hospital	psychosocial work factors, psychological distress, burnout, sleeping problems
Bryan et al. (1998)	[45]	United States	Nurses	Hospital	job satisfaction
Bunce and West (1996)	[46]	United Kingdom	Mixed	Mixed	job motivation, job satisfaction, psychological strain, job-induced tension
Carson et al. (1999)	[47]	United Kingdom	Nurses	Hospital	occupational stressors, psychological distress, burnout
Doran et al. (2015)	[48]	Canada	Nurses	Mixed	work environment, organizational commitment and job satisfaction, burnout
Ewers et al. (2002)	[49]	United Kingdom	Nurses	Mental Health Unit	burnout
Finnema et al. (2005)	[50]	The Netherlands	Nurses	Nursing home	stress, stress reactions, work satisfaction, absenteeism
Ghazavi et al. (2010)	[51]	Iran	Nurses	Hospital	occupational stress
Gregory et al. (2018)	[52]	United States	Physicians	Primary care clinics	burnout
Gunusen and Ustun (2010)	[53]	Turkey	Nurses	Hospital	burnout
Haggstrom et al. (2005)	[54]	Sweden	Nurses	Nursing home	work satisfaction and dissatisfaction
Hall et al. (2008)	[55]	Canada	Nurses	Hospital	satisfaction, stress, work environment, role tension
Heaney et al. (1995)	[56]	United States	Care staff and managers	Residential care facility	social support, organizational climate for participation and influence in decision-making, employees’ confidence in ability to cope with common work problems, psychological wellbeing
Hyman (1993)	[57]	United States	Mixed	Residential care facility	burnout, work atmosphere
Jeon et al. (2015)	[58]	Australia	Mixed	Residential and community aged care sites	work environment, staff turnover, stress, absenteeism
Joyce et al. (2011)	[59]	Australia	Nurses	Hospital and community health settings	mental health literacy for peer support
Kapoor et al. (2018)	[60]	United States	Mixed	Hospital	burnout, grief, distress
Koivu et al. (2012)	[61]	Finland	Nurses	Hospital	psychological and social factors at work, burnout, psychological distress
Lavoie-Tremblay et al. (2005)	[62]	Canada	Mixed	Hospital	decision latitude, psychological demands, social support, effort/reward imbalance, reward, psychological distress, absenteeism
Le Blanc et al. (2007)	[63]	The Netherlands	Mixed	Hospital	burnout, social support, participation in decision making, job control, job demands
Ledikwe et al. (2018)	[64]	Botswana	Mixed	Public health facilities	job satisfaction, psychological wellbeing, burnout, stress
Leiter et al. (2011 and 2012); Oore et al. (2010)	[65,66,67]	Canada	Mixed	Hospital	workload, job control, incivility, respect, negative affect and anxiety
Linzer et al. (2015 and 2017)	[68,69]	United States	Physicians	Primary care clinics	work control, stress, burnout, chaos, likelihood to leave
Loiselle et al. (2012)	[70]	Canada	Nurses	Hospital	performance obstacles, perceived work support, emotional distress
Lucas et al. (2012)	[71]	United States	Physicians	Hospital	burnout, stress, workplace control
McDonald et al. (2012 and 2013)	[72,73]	Australia	Mixed	Hospital	personal resilience
Melchior et al. (1996)	[74]	The Netherlands	Nurses	Hospital	burnout
Mikkelsen et al. (2000)	[75]	Norway	Mixed	Community health care	work-related stress, subjective health, demands-control, social support, role harmony,
Newman et al. (2015)	[76]	Australia	Nurses	Correctional centers and forensic health	burnout, workplace satisfaction
Odle-Dusseau et al. (2016)	[77]	United States	Supervisors	Nursing home	job satisfaction, organizational commitment, turnover intentions, employee engagement, work-family conflict
Parsons et al. (2004)	[78]	United States	Nurses	Hospital	control, work satisfaction, interactions, organizational commitment
Peterson et al. (2008)	[79]	Sweden	Mixed	Hospital	burnout, quantitative demands, anxiety, depression, general health
Petterson and Arnetz (1998)	[80]	Sweden	Mixed	Hospital	job demands, work pressure, psychosomatic symptoms, exhaustion, job control, coping
Petterson et al. (2006)	[81]	Sweden	Nurses	Elder care organizations	work demands, job control, support, psychosomatic symptoms, stress, coping
Proctor et al. (1998)	[82]	United Kingdom	Elder care staff	Nursing and residential homes (elder care)	general health, occupational stress
Quenot et al. (2012)	[83]	France	Mixed	Hospital	burnout, depression
Razavi et al. (1993)	[84]	Belgium, France	Nurses	Hospital	occupational stress, attitudes
Redhead et al. (2011)	[85]	United Kingdom	Nurses	Inpatient care for mental health patients	burnout
Richter et al. (2014)	[86]	Germany	Physicians	Hospital	burnout, physical and mental health
Rickard et al. (2012)	[87]	Australia	Nurses	Hospital	occupational stress, turnover
Romig et al. (2012)	[88]	United States	Nurses	Hospital	psychological working conditions, burnout, relations and communications
Saint-Louis and Bourjolly (2018)	[89]	United States	Mixed	Oncology Unit	N/A
Schrijnemaekers et al. (2003)	[90]	The Netherlands	Elder care staff	Residential care facility	job satisfaction, burnout, sick leave
Sexton et al. (2014)	[91]	United States	Mixed	NICUs	burnout
Spetz (2008)	[92]	United States	Nurses	Hospital	job satisfaction
Takizawa et al. (2017)	[93]	Japan	Mixed	Elder care organizations	job stress, coping
Traeger et al. (2013)	[94]	United States	Mixed	Oncology Unit	burnout, stress
Uchiyama et al. (2013)	[95]	Japan	Nurses	Hospital	depression, psychosocial work environment
Van Bogaert et al. (2014)	[96]	Belgium	Mixed	Hospital	job satisfaction, burnout, intent to leave
Wallbank (2010)	[97]	United Kingdom	Mixed	Hospital	burnout, compassion satisfaction, compassion fatigue, stress
Wei et al. (2017)	[98]	China	Nurses	Hospital	burnout
Yamagishi et al. (2007)	[99]	Japan	Nurses	Hospital	job stress, depression

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
