# Peer review of "Workplace-Based Organizational Interventions Promoting Mental Health and Happiness among Healthcare Workers: A Realist Review"

_ijerph, 2019, doi:10.3390/ijerph16224396_

Round 1
Reviewer 1 Report
This is a very well conducted study. It brings relevant contribution for both research and practice.
Regarding the manuscript ijerph-617566, I can highlight some strength and weakness:
- This is a very well-conducted realistic review about a very relevant issue in occupational health;
- The authors synthesized the evidence in a very practical way, providing important clues for those engaged in occupational health services and research;
- Although I agree that mental health needs to be approached in a more positive view I do not agree about the inclusion of the term "happiness" in the title, abstract and introduction because most of the studies did not assess happiness as an outcome;
- The PRISMA guideline could be followed by authors;
- Table S1 could be shortened and included in the main text.
Author Response
We would like to thank the reviewer for their thoughtful comments and feedback on our realist review. In response to the reviewer’s specific comments:
This is a very well-conducted realistic review about a very relevant issue in occupational health.
We appreciate the reviewer’s feedback that the article provides a relevant contribution to an important issue in occupational health – the mental health of healthcare workers.
The authors synthesized the evidence in a very practical way, providing important clues for those engaged in occupational health services and research.
We appreciate the reviewer’s positive feedback on the practicality of the article.
Although I agree that mental health needs to be approached in a more positive view I do not agree about the inclusion of the term "happiness" in the title, abstract and introduction because most of the studies did not assess happiness as an outcome.
In this article, we use the term “happiness” to broadly encompass mental wellbeing which is in line with such usage in the field of economics and the growing academic interest around concepts of positive mental wellbeing. For example, and as noted in lines 420 – 422 of the manuscript, the World Happiness Report – an annual report published by the United Nations since 2012 – uses life satisfaction as its measure of happiness, indicating the broad conceptualization of “happiness” beyond emotional happiness. Further, in lines 411-413, we highlight that the term “happiness” can indeed refer both to moods and emotions as well as to more long-term wellbeing and life satisfaction. As such, we intentionally used the term “happiness” to capture and denote mental wellbeing more broadly. In terms of the integration of the concept of “happiness” in the review’s methods, we included positive constructs of mental health including happiness in our literature search strategy (as listed in the Appendix); however, as discussed in the Discussion and Conclusion sections, there remains an opportunity to integrate positive mental health and wellbeing constructs in the context of workers’ mental health. Noting this reviewer’s attention to this issue, we have made our final sentence more explicit as to the need for more studies:
“As well, there exists great opportunity for better integrating use of positive mental health and wellbeing constructs such as happiness in the context of workers’ mental health in intervention studies, so that efforts to improve this can be more strongly considered in future knowledge synthesis reviews.”
The PRISMA guideline could be followed by authors.
We have reduced Table S1 and included in the main text as Table 2, following the PRISMA diagram so that the authors are listed in this location.
Table S1 could be shortened and included in the main text.
As noted above, we have reduced Table S1 and included in the main text as Table 2. We have left the full version of Table S1 as a supplemental material.
Reviewer 2 Report
This is an excellent review adding to the literature in terms of pulling together and making sense of an enormous amount of articles in the area.
Findings appreciated and whilst indicating the need for more work, also already providing direction for (further) application. Thoughtful interpretation which should direct some of that further research.
Extremely well written. Apart from a small amount to of tautology, I found the write up to be commendably clear.
A small point: not including any grey literature could have been a limitation, for there has been some commissioned research to understand what is going on "on the ground" by enforcement agencies such as UK HSE (eg Beacons of Excellence in Stress Prevention".
Reference 37 has repeated title.
Author Response
We appreciate the reviewer’s positive feedback including on the article’s quality and value to the area of mental health promotion among healthcare workers.
In response to the two specific items noted by the reviewer:
We have added the exclusion of grey literature as an additional limitation in the “Limitations” section, and included reference to the “Beacons of Excellence in Stress Prevention” study as an example of what could then be considered.
We have removed the second title from reference 37.